# Automated mold defects classification in paintings: A comparison of machine learning and rule-based techniques

Hilman Nordin[1,2], Bushroa Abdul Razak[1,3], Norrima Mokhtar[4]*, Mohd Fadzil Jamaludin[3], Adeel Mehmood[5]

1 Faculty of Engineering, Department of Mechanical Engineering, Universiti Malaya, Kuala Lumpur, Malaysia, 2 Sunway Centre for Digital Humanities and Cultural Heritage, Sunway University, Sunway City, Selangor, Malaysia, 3 Faculty of Engineering, Centre of Advanced Manufacturing and Material Processing (AMMP Centre), Universiti Malaya, Kuala Lumpur, Malaysia, 4 Faculty of Engineering, Department of Electrical Engineering, Universiti Malaya, Kuala Lumpur, Malaysia, 5 Faculty of Science and Engineering, School of Computer Science, University of Hull, Hull, United Kingdom

☯ These authors contributed equally to this work.
* norrimamokhtar@um.edu.my

**Data Availability Statement:** All dataset files is available in GitHub repository which can be accessed directly using this link: (https://github.com/resigned4life/ml-mold-classification.git).

## Abstract

Mold defects pose a significant risk to the preservation of valuable fine art paintings, typically arising from fungal growth in humid environments. This paper presents a novel approach for detecting and categorizing mold defects in fine art paintings. The technique leverages a feature extraction method called Derivative Level Thresholding to pinpoint suspicious regions within an image. Subsequently, these regions are classified as mold defects using either morphological filtering or machine learning models such as Classification and Regression Trees (CART) and Linear Discriminant Analysis (LDA). The efficacy of these methods was evaluated using the Mold Features Dataset (MFD) and a separate set of test images. Results indicate that both methods improve the accuracy and precision of mold defect detection compared to no classifier. However, the CART algorithm exhibits superior performance, increasing precision by 32% to 53% while maintaining high accuracy (96%) even with an imbalanced dataset. This innovative method has the potential to transform the approach to managing mold defects in fine art paintings by offering a more precise and efficient means of identification. By enabling early detection of mold defects, this method can play a crucial role in safeguarding these invaluable artworks for future generations.

## Introduction

Fine art has long been hailed as the pinnacle of human creativity in past eras. Through visual presentation of ideas and concepts, viewers are drawn into the exploration of meaning within artwork, offering a unique engagement compared to textual presentations. This distinctive method of conveying concepts has been refined over time, giving rise to esteemed fine art masters whose works are still highly regarded today. These artworks hold significant value, with some commanding high prices far exceeding material costs involved in their creation.

**Funding:** The author(s) received no specific funding for this work.

**Competing interests:** The authors have declared that no competing interests exist.

Paintings, as a form of art, encapsulate human expression and intellect, serving as reflections of cultural, historical, social, and political contexts. It is these qualities that have endowed paintings with enduring value, making them an important part of heritage today and earning artists recognition during the Renaissance period in Europe.

Paintings, being historical artifacts, require meticulous care due to various factors that can impact their preservation. Fine art paintings, particularly older ones, are susceptible to deterioration over time from influences such as exposure to light, humidity, pollutants, and improper handling. These defects not only compromise the aesthetics of the paintings but also diminish their intrinsic properties and integrity, risking their permanent loss. Airborne pollutants can trigger chemical reactions on artwork surfaces, while artwork materials such as pigments and adhesives can contribute to degradation [1]. Biological agents such as insects, fungi and micro-organisms can also attach themselves to organic materials like wood frames, canvas or paper, further jeopardizing the artwork's integrity [1]. Safeguarding paintings from these threats is crucial, as is early detection to facilitate appropriate corrective or preventive actions [2].

Detecting defects in paintings is a crucial step in the restoration, preservation, and conservation of art. Some of the types of defects on paintings include cracks, molds, blistering, sagging, and discoloration. Mold infestations, for example, are prevalent in paintings stored in humid environments where moisture fosters mold growth under favorable conditions. The presence of red-brownish spots on paintings, commonly referred to as "foxing," is a recognized fungal occurrence resulting from the growth of bacteria or mold on paper [1].

To address this issue, art conservators would conduct a thorough visual inspection of the painting to identify any visible indication of mold growth. This typically involves utilizing various forms of illumination and tools such as microscopes or magnifying glasses [2]. Visible changes in the painting's appearance such as discolored patches, blurry specks, and powdery stains would be identified to confirm the presence of mold growth. Detecting mold growth in its early stages is possible but presents a significant challenge when dealing with large artworks. The task demands intense focus, drawing on the conservators' expertise and experience to differentiate between actual defects and inherent characteristics of the cultural artifacts [3]. Understanding the painting's history of creation, materials used, and the type of defects is essential for implementing appropriate corrective actions to preserve its condition [2,4]. Furthermore, documenting the skills and judgement of experts is vital, as justifying intervention requires evaluating not only the physical state of the cultural heritage, but also the level of repairs necessitating expert assessment [3]. By documenting these aspects, the expertise and judgment of conservators can also be shared more widely, enhancing accessibility to art restoration practices.

The advancement of technology has paved the way for artificial intelligence (AI) to emulate human capability to a certain degree, particularly in specific tasks such as classification. By leveraging algorithms and machine learning techniques, AI can streamline routine and focus-heavy tasks such as inspection, detection, and categorization of defects. Employing the right methodologies can enhance conservation efforts by complementing the work of art restorers and offering insights such as defect sizes and distributions, and other swiftly calculable metrics. Essentially, this introduces a computer assistant to the art restoration experts, capable of learning from the expert and improving its proficiency over time through machine learning algorithms [5].

Machine learning applications in fine art painting defect classification remain limited, yet the technology has found extensive use in defect detection across various domains. By effectively categorizing target images into background and defects, machine learning systems can identify different types of defects, ranging from indications of disease in biological samples [6–9], cracks and holes in architectural structures [10–13], to anomalies on manufactured surfaces [5,14–23].

Machine learning has expanded beyond defect identification and classification. It is now used to regulate factors influencing defect formation in processes like laser hot wire cladding [24], aerospace weld inspection [25] and fruit quality assessment [26]. Other applications include predictive maintenance [27], fatigue-life prediction [28], process modeling in additive manufacturing [29], online anomaly detection [30], equipment corrosion risk assessment [31], and assembly defect recognition [32].

While the application of machine learning to fine art painting defects is still relatively unexplored, the potential for using these techniques to identify and classify defects in paintings is promising. By leveraging advancements in machine learning and computer vision, it may be possible to use the technology to assist art conservators in identifying and addressing defects in valuable artworks.

## Related works

Given the current limitation in defect detection in fine art paintings, a practical strategy involves utilizing surface defect detection methodologies, considering paintings primarily consist of painted surfaces that may exhibit imperfections. Within this context, defect classification entails the categorization of identified elements as either confirm defects or background noise. In the field of surface defect detection, apart from employing sensors operating in the visible spectrum, methods such as thermography [16,33,34], computed tomography (CT) scanning [35], eddy current testing [36], guided wave signals [37], acoustic emission [38], and 3D scanning [39] are utilized to quantify the defect information on the surface.

The process of categorizing defect candidates into defects and background can be achieved through two primary methods. The first method utilizes rule-based techniques, where a predefined set of criteria or rules is applied to classify the candidates. The second method involves the use of machine learning models that are trained on extracted features to perform the classification tasks. Rule-based methods typically employ a fixed algorithm tailored for specific cases, providing an advantage in those scenarios. However, in recent years, machine learning has gained popularity in surface defect detection and classification.

Rule-based methods continue to hold a pivotal role in the feature extraction process, forming the basis for the initial identification of defects. One method used for feature extraction for surface defects on fine art painting is the Derivative Level Thresholding algorithm [40]. These extracted features are manually categorized to create the ground truth image, which will then be used for the training the machine learning models [14,22,23,31,32,41]. In this study, the machine learning model will focus solely on the classification of mold defects. Incorporating machine learning into the classification problem has been shown to enhance the accuracy of defect detection significantly.

Although both rule-based methods and machine learning offer distinct advantages, their effectiveness can vary depending on the specific characteristics of the defect detection task. Rule-based methods excel in well-defined scenarios and controlled environments where the criteria for defect classification are clear and unambiguous. They are transparent and easier to interpret, making it easy to understand how the system arrives at its decisions. However, they may struggle to adapt to new or unexpected defect patterns. Moreover, developing and maintaining a comprehensive set of rules can be time-consuming.

Machine learning models, on the other hand, can learn from diverse datasets and adapt to varying defect characteristics, making them more robust and flexible. They reduce the need for manual rule creation and maintenance, so they are able to handle complex tasks that would be difficult to define with rules. However, they require large amounts of high-quality data to

train, and the decision-making process can be opaque, making it difficult to understand how they arrive at their predictions.

The quantity and quality of training data are crucial for the accuracy of machine learning predictions. Various methods can be employed to enhance the training datasets, including dividing high-resolution data into smaller segments, generating artificial data, creating defects in experimental results, and utilizing publicly available data. Segmenting high-resolution data into smaller parts can streamline processing, potentially boosting training speed [10,14]. In some cases, when actual data collection is limited, artificially generated data can represent real-world scenarios, aiding in dataset creation [16,25,36]. In cases where the defects are rare but significant, artificially creating defects in experimental results can be considered. [37,42,43]. The accuracy of machine learning predictions can also be enhanced by enriching the datasets by leveraging publicly available data [5,6,18,41].

The importance of defect detection and classification in fine art preservation is paramount for conserving cultural heritage. Surface defects on paintings require meticulous treatments, and these defect detection and classification methods aid art restorers in assessing damage levels accurately. While existing methods that leverage on machine learning have tackled various challenges in defect detection in many industries, further research is still required, especially in the realm of cultural heritage artifacts. To address this need, this study compares rule-based and machine learning filtering for mold defect classification in fine art painting. The proposed framework deviates from the Derivative Level Thresholding (DLT) algorithm [40] by employing a machine learning defect classifier trained on segmented high-resolution painting images, facilitating training and evaluation processes.

## Methodology and testing

### Mold defect detection and classification method

In this paper, we address the challenging task of mold defect classification in paintings. To tackle this problem, we compare two different approaches: a rule-based method called morphological filtering and machine learning methods. Both approaches involve utilizing the Derivative Level Thresholding (DLT) technique to extract potential defects from paintings, generating a binary image that highlights the defects in spatial relation to one another. Subsequently, the two approaches are applied to classify these potential defects as either molds or intrinsic background elements, effectively distinguishing defects from the painting itself. Fig 1 illustrates the methodology for detecting and classifying surface defects, specifically mold defects, on fine art paintings.

### Preliminary defect location

Derivative Level Thresholding (DLT) is a technique that utilizes derivative operators to segment images based on the rate of change of pixel intensities. This approach can be used as an efficient mold defect identification tool, transforming input images into binary output representations. Initially, sample images are converted into grayscale before undergoing multi-level thresholding processes to generate binary images. An optimal binary image can then be identified from the analysis of subtracted binary images. The DLT algorithm effectively isolates relevant data from this optimal binary image by suppressing objects that are not mold defects, thereby enhancing defect detection accuracy by minimizing noise interference.

Fig 2 demonstrates the effectiveness of the DLT method in segmenting defects. It features a set of selected sample images, along with their corresponding ground truth images, highlighting the results of DLT binarization. This process generates a binary image that can be directly

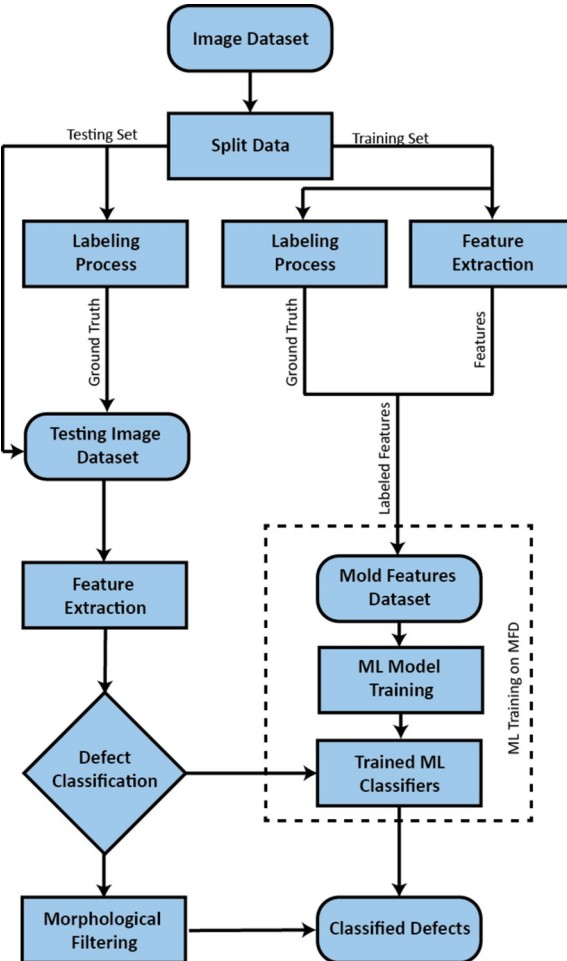

**Fig 1. Process overview flow chart.**

utilized for defect detection, enabling the accurate identification of mold within the sample images.

In this study, a classification methodology is proposed to improve accuracy, as the current resulting image contains considerable noise. The approach involves applying a connected component process to each object on the binary images using the flood fill algorithm. This technique groups the objects into distinct blobs, which represent individual defect candidates. Furthermore, the process extracts various features of the defect candidates, which can then be processed for classification. This classification step enables noise removal, thereby enhancing the overall accuracy of the image.

## Mold features dataset

An image dataset was compiled from 566 high-resolution sample images of two fine art paintings [41]. These images were captured using a high-resolution scanner (Niji-X) equipped with a line sensor operating within the visible light spectrum, ensuring the capture of essential color details crucial for fine art analysis. Each sample image is sized at 256 x 256 pixels with a resolution of 650 dpi, corresponding to a physical area of 10mm x 10mm. Human evaluators then assessed the samples to differentiate genuine mold defects from inherent panting features,

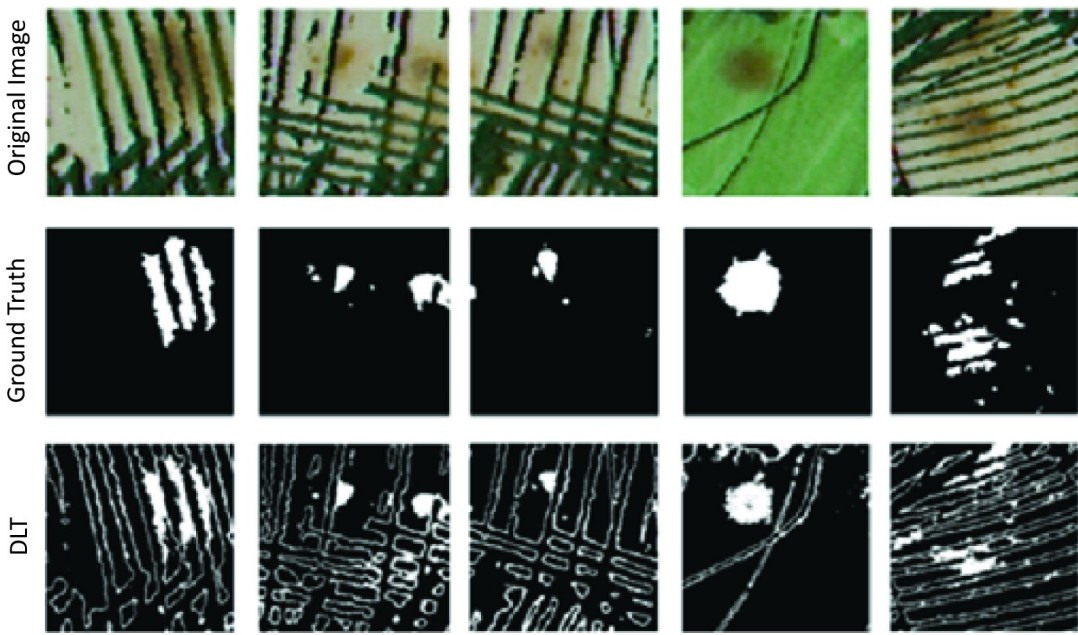

**Fig 2. Sample binary images produced using the DLT feature extraction method.**

creating corresponding ground truth image for each original sample. From this dataset, 100 images were chosen and categorized into training, validation, or testing sets. 80 images were selected from the initial 100 images and grouped as training images. These images underwent a features extraction process, where 10 features were identified from the morphology of each defective candidate. These features include region area, major axis length, minor axis length, eccentricity, filled area, extent, perimeter, equivalent diameter, convex area, and solidity. Each defective candidate's features were compared to the corresponding ground truth image and labelled as molds (MOLD) or non-molds (NO-MOLD). Hence, the resulting database contained 11 headings, which included the 10 features and an additional heading for the classification of each defect candidate.

Table 1 displays the characteristics of each of the features in the Mold Features Dataset (MFD), including statistical descriptions such as mean, standard deviation, minimum and maximum values for the overall dataset of the mold defect candidates. The 80 training images

**Table 1. Distribution of quantitative mold features used in the classification models.**

| Feature | Name | Mean | Std | Min | Max |
|---|---|---|---|---|---|
| F1 | Region Area (pixel area) | 36.50 | 575.94 | 1 | 51503 |
| F2 | Major Axis Length (pixel length) | 7.79 | 19 | 1.15 | 377.29 |
| F3 | Minor Axis Length (pixel length) | 2.75 | 6.75 | 1.15 | 304 |
| F4 | Eccentricity | 0.60 | 0.43 | 0 | 1 |
| F5 | Filled Area (pixel area) | 42.73 | 661.56 | 1.00 | 52329 |
| F6 | Extent | 0.81 | 0.27 | 0.02 | 1 |
| F7 | Perimeter (pixel length) | 18.35 | 84.88 | 0 | 3677.51 |
| F8 | Equivalent Diameter (pixel length) | 3.28 | 5.98 | 1.13 | 256.08 |
| F9 | Convex Area (pixel area) | 87.82 | 1161.01 | 1.00 | 65536 |
| F10 | Solidity | 0.90 | 0.18 | 0.07 | 1 |

contained 30,868 defect candidates, with 757 assigned as "MOLD" and 30,111 assigned as "NO-MOLD" for the MFD. The dataset has a significant class imbalance, with only 2.5% of the data attributed to mold defects, making it a challenge for machine learning algorithms to predict the minority class.

Apart from being used for training, the remaining images in the MFD are designated for validation and testing, with ten images assigned to each. The ten images allocated for testing are also specifically utilized to evaluate the effectiveness of the morphological filtering mold defect classification method, as detailed in the subsequent section.

## Mold defect classification: Morphological filtering

The morphological filtering method used in this study is an example of a rule-based algorithm. Rule-based algorithms are distinguished by their direct and specific nature, in contrast to the more general task-oriented approach of machine learning that requires training data with human-labeled ground truth [20]. This method typically involves statistical analysis, and good results can be achieved by implementing multiple layers of different algorithms. Specifically, the morphological filtering process in this study consists of three distinct filter layers: size filter, hole filter and line filter, applied to each defect candidates. The filtered image, devoid of noise, is generated following the filtering process and is then compared to the ground truth image for evaluation. The morphological filtering process involving the three filter layers is detailed as follows:

**Size filter.**  Firstly, the defect candidates are filtered according to size. The size filter will store pixel groups that are larger than $a_{min}$ and smaller than $a_{max}$:

$$a_{min} < A_k < a_{max} \tag{1}$$

Single pixels and groups of pixels that are connected diagonally will not be stored. The value of $a_{min}$ is set to 20, while $a_{max}$ is set to 5000 pixels. The selected group of pixels will then be refined using the hole filter.

**Hole filter.**  The hole filter works by filtering groups of pixels which have a difference between the filled image area and the image area, $A_k$, larger than a preset scale threshold value. A filled image area is the total number of pixels in the same group of pixels with holes filled. By using this hole filter, groups of pixels with holes will be filtered out. In this study, the hole filter threshold value is set at 0.1.

The filter is calculated by:

$$\frac{A_{filled} - A_k}{A_k} > 0.1 \tag{2}$$

**Line filter.**  Finally, a line filter will be applied to the defect candidate images. In this process, the pixel group is tested to see if it has an area-to-perimeter ratio of more than 1.3. This filter will eliminate lines, which usually have a low area-to-perimeter ratio, especially for straight lines.

$$\frac{A_k}{Perimeter_k} > 1.3 \tag{3}$$

In the context of morphological filtering, since training and validation were not required, only the ten sample images from the MFD testing group are directly utilized to evaluate the mold defect classification method.

## Mold defect classification: Machine learning

Machine learning classification offers the advantage of leveraging machines to learn from labelled features, thereby reducing the need for human intervention in discerning the statistical aspects of the dataset classification. The classification performance improves as more data is progressively introduced to the system. In this study, several machine learning models were implemented, including Logistic Regression (LR), Linear Discriminant Analysis (LDA), K-Nearest Neighbors (KNN), Classification and Regression Trees (CART), Gaussian Naïve Bayes (NB), Random Forest (RF), Support Vector Machines (SVM) and Extreme Gradient Boosting (XGB). While LR and LDA are linear algorithms, the rest are non-linear.

To evaluate these algorithms, a validation process was established using the Mold Features Dataset (MFD). This validation group served as the hold-out data, enabling the evaluation of model performance before finalizing the choice of model. The MFD dataset was divided into two parts, with 80% utilized for training, evaluation, and selection of the optimal model, while the remaining 20% was reserved as the validation dataset. To achieve this division, the train_-test_split() function from the scikit-learn library was utilized, ensuring a consistent test size of 0.20 and maintaining a fixed split using random_state = 1. This standardization is important to enable the direct comparison and evaluation of the model using the same data split, which is necessary for fair assessment. To estimate the model's accuracy, a test harness was established using stratified 10-fold cross validation, which involves training the models on nine parts and testing them on one part out of the 10-part split. This approach ensures that each fold maintains the same class distribution as the MFD. The models were then evaluated using the cross_-val_score() function, with the scoring parameter set to *scoring = 'accuracy'*.

A comparison was conducted on the different machine learning algorithms. The best fit was modeled and trained on the dataset for prediction on unseen data. The model classifies each line of the dataset containing all the pixel groups into either "MOLD" or "NOMOLD" categories. By turning ON or OFF the pixel groups based on the classification, a binary image is produced that only contains pixel groups classified as "MOLD". The machine learning models were coded, trained and evaluated using Python 3.9.12 with SciPy 1.7.3, Numpy 1.21.5 and Scikit-Learn 1.0.2 libraries in Spyder 5.1.5 Integrated Development Environment.

## Evaluation metrics

To evaluate the methods discussed in this study, several metrics were selected to assess their performance. The resulting images from these methods were compared with the ground truth image to calculate the metrics. Two sets of metrics were employed in this study. The first set includes metrics commonly used in general tests for identifying true positives, such as accuracy, sensitivity, and precision.

The second set of metrics is specifically used in the field of machine learning, as they are more robust in determining the performance of a model or network in binary classification. Some of the metrics are typically used to measure the efficiency of a model in the presence of an imbalanced dataset, which is the case in this study. The metrics include Intersection over Union (IoU), F1-measure and Matthews Correlation Coefficient (MCC).

**Accuracy.** The accuracy metric accounts for the correct classification of both mold defects and the background. It can be calculated using the formula:

$$\text{Accuracy} = \frac{TP + TN}{TP + FP + FN + TN} \tag{4}$$

where TP represents True Positive, FP represents False Positive, FN represents False Negative and TN represents True Negative. These metrics consider the value given by the ratio of

correctly classified mold defects and the background to the distribution, which is the total number of pixels in the sample image.

**Precision.** Precision can be calculated using the formula:

$$Precision = \frac{TP}{TP + FP} \tag{5}$$

where TP is True Positive and FP is False Positive.

**Sensitivity.** The sensitivity, also referred to as recall or true positive rate, is determined using the formula:

$$Sensitivity = \frac{TP}{TP + FN} \tag{6}$$

where TP represents True Positive, and FN represents False Negative. This metric measures the recall of the evaluated methods. The value is determined by the ratio of the pixels correctly classified mold defects by the methods to the overall pixel groups classified as molds in the ground truth image.

**Intersection over Union (IoU).** The Intersection Over Union (IoU), also referred to as the Jaccard's Index, is a precise measure commonly used to assess semantic segmentation models. It is calculated as the ratio of the overlapping area between the A (predicted mold pixels) and B (ground truth mold pixels) over the combined area of both A and B.

$$IoU = \frac{Overlapping\ Area}{Combined\ Area} = \frac{TP}{TP + FP + FN} \tag{7}$$

**F1-measure.** The F1-measure is calculated by:

$$F1_{score} = \frac{2TP}{2TP + FP + FN} = \frac{Precision \times Recall}{Precision + Recall} \tag{8}$$

where TP represents True Positive, FP represents False Positive, and FN represents False Negative. The F1-measure, also referred to as the Dice coefficient, is recognized as the harmonic mean of precision and recall.

**Matthews Correlation Coefficient (MCC.** The Matthews Correlation Coefficient (MCC) is a metric that evaluates the performance of a model or network by considering all four quadrants of the confusion matrix. This characteristic makes it a dependable statistical metric suitable for measuring classification performance [44].

$$MCC = \frac{TP.TN - FP.FN}{\sqrt{(TP + FP).(TP + FN).(TN + FP).(TN + FN)}} \tag{9}$$

## Results and discussions

In this section, the results are presented with a comparison of two different methods. The first method is a basic morphological filtering using connected component analysis on the DLT binarized images, where the classification is based on whether the group of pixel groups on the binary image conform to a set of filters. The second method uses machine learning models to accurately classify pixel groups from the DLT threshold images, trained on the MFD dataset. The comparative performance of the two methods for defect classification is then evaluated in detecting mold defects on the selected ink-on-paper sketch artwork. The testing dataset

**Table 2. Performance of the machine learning models on MFD.**

| Machine Learning Model | Training Performance | | | Testing Performance | | | |
| --- | --- | --- | --- | --- | --- | --- | --- |
| | Accuracy | Standard Deviation | Time Taken | Detection | Sensitivity | Precision | Accuracy |
| Linear Discriminant Analysis (LDA) | 0.967685 | 0.003112 | 0.545020 | 100% | 0.615854 | 0.369191 | 0.930495 |
| Gaussian Naïve Bayes (NB) | 0.959545 | 0.002016 | 0.284603 | 90% | 0.572442 | 0.273448 | 0.921675 |
| Classification and Regression Trees (CART) | 0.966429 | 0.002241 | 0.969686 | 80% | 0.428373 | 0.421317 | 0.954149 |
| Extreme Gradient Boosting (XGB) | 0.970681 | 0.001928 | 115.716518 | 80% | 0.322591 | 0.200445 | 0.925653 |
| Random Forest (RF) | 0.975136 | 0.001482 | 4.445949 | 70% | 0.377082 | 0.518940 | 0.956317 |
| K-Nearest Neighbors (KNN) | 0.974326 | 0.001907 | 2.271064 | 50% | 0.159770 | 0.417230 | 0.955942 |
| Support Vector Machine (SVM) | 0.940147 | 0.005339 | 96.929111 | 0 | 0 | - | 0.957342 |
| Logistic Regression (LR) | 0.975460 | 0.000994 | 2.541826 | 0 | 0 | - | 0.953615 |

contains ten samples selected from various locations on the scanned image of the artwork, with known mold location from the paintings. The respective ground truth images in the dataset are used to compare the performance of each mold defect classification method.

## Performance comparison for machine learning classifiers

Table 2 shows the performance of the models based on the validation set for all machine learning algorithms considered in this study. It provides the accuracy of each algorithm along with the corresponding standard deviation and time taken for the training, offering insights into the classification performance of these methods. The table also includes the evaluations of each of the machine learning algorithms on the testing dataset. Additionally, Fig 3 presents a box and whisker plot on the training performance for each of the evaluated algorithms.

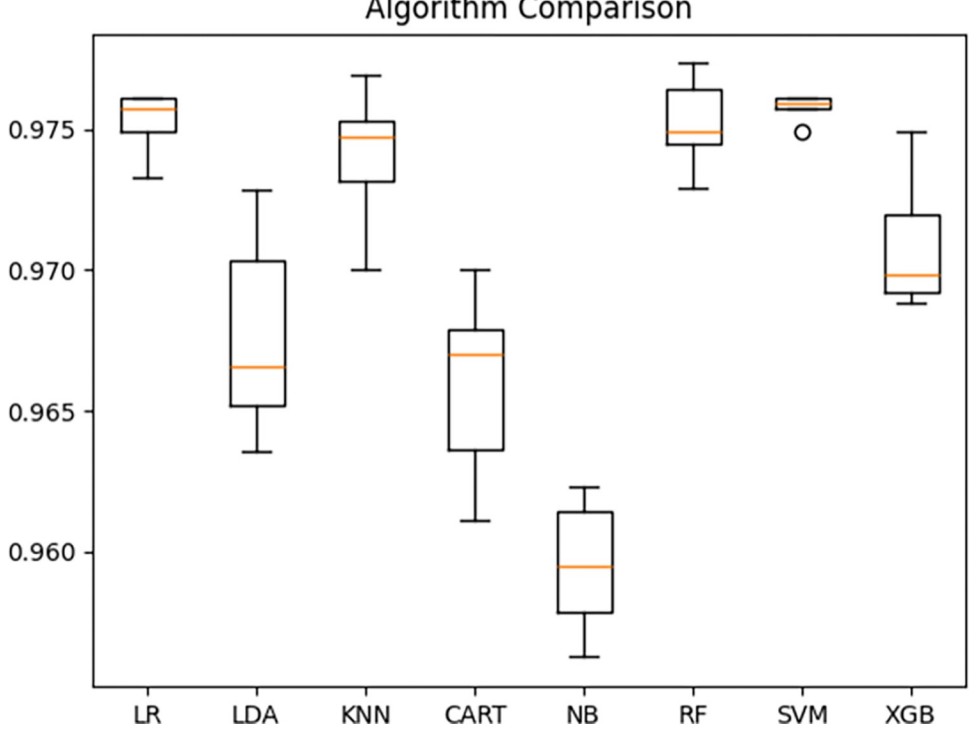

**Fig 3. Machine learning algorithm validation performance comparison plot.**

In Fig 3, it is evident that the LR, KNN and SVM algorithms exhibited more consistent and accurate predictions in the validation set, as indicated by the smaller box in the boxplot. This suggests that the predictions generated by these algorithms were closely clustered around the median. However, the Liblinear algorithm used for training the models encountered difficulty in solving the optimization problem after multiple iterations, resulting in the "Liblinear failed to converge" error for LR. As shown in Table 2, both LR and SVM were unable to detect any defects in the testing dataset, despite the absence of convergence errors for SVM. This issue stems from the imbalance in the dataset, which leads to high accuracy values even when no defects are detected.

These results revealed limitations in the defect classification capabilities of certain methods. For instance, KNN achieved high training (97.4%) and testing accuracy (95.6%), However, this performance stemmed primarily from accurately classifying the background, as evidenced by the low sensitivity value (16%). Notably, KNN only identified defects in 5 out of the 10 test images. For a comparative analysis with morphological filtering, we selected machine learning classifiers (LDA, CART and NB) based on their high accuracy (92%– 97%) and their ability to correctly identify defects in at least 8 out of 10 images. The two test images which do not have any detected defects from CART were excluded from further analysis.

## Qualitative comparison: Machine learning vs. morphological filtering

The performance of the mold defect classification methods selected for this study is based on the previously discussed metrics. Figs 4 and 5 illustrate the performance of the two binary defect classification methods across 8 testing image samples from the testing set, providing a comprehensive overview of the performance of five different methodologies. The first source comprises of evaluation metrics of binary images directly obtained from the Derivative Level Thresholding method, labelled as DLT in the subsequent figures. The second source is from morphological filtering method, denoted as MF, while the third, fourth and fifth are from LDA, CART and NB machine learning classification, respectively.

Both rule-based and machine learning classification methods generally improved accuracy, precision, F1-score, IoU and MCC compared to the original DLT method. Notably, all classification methods exhibited a decrease in sensitivity due to false negatives. MF achieved the highest accuracy and precision, outperforming all machine learning methods except for CART. CART demonstrated superior accuracy, precision, F1-score and MCC, but the lowest sensitivity. Conversely, LDA and NB achieved higher sensitivity than MF. This suggests that machine learning excels in sensitivity, a crucial metric for imbalanced dataset like the MFD. Overall, applying filters (morphological or machine learning) significantly improved all metrics compared to the DLT method alone. This highlights the benefit of classification in enhancing defect detection performance.

Additionally, we conducted a significance analysis, similar to [45], to compare the performance of LDA, NB, CART and MF against the original defect detection method without classification using paired t-tests. The paired t-test revealed significant t-statistics and p-values for all methods, as shown in Table 3. Overall, CART emerged as the top performer based on mean accuracy and standard deviation. However, all four classifiers demonstrated significant improvements over the original method, highlighting the benefits of classification. While MF achieved lower mean accuracy than CART, it outperformed LDA and NB. This suggests that MF, though not as accurate as CART, is a competitive option, especially when interpretability and simplicity are priorities. The results emphasize the effectiveness of machine learning methods for this dataset and task.

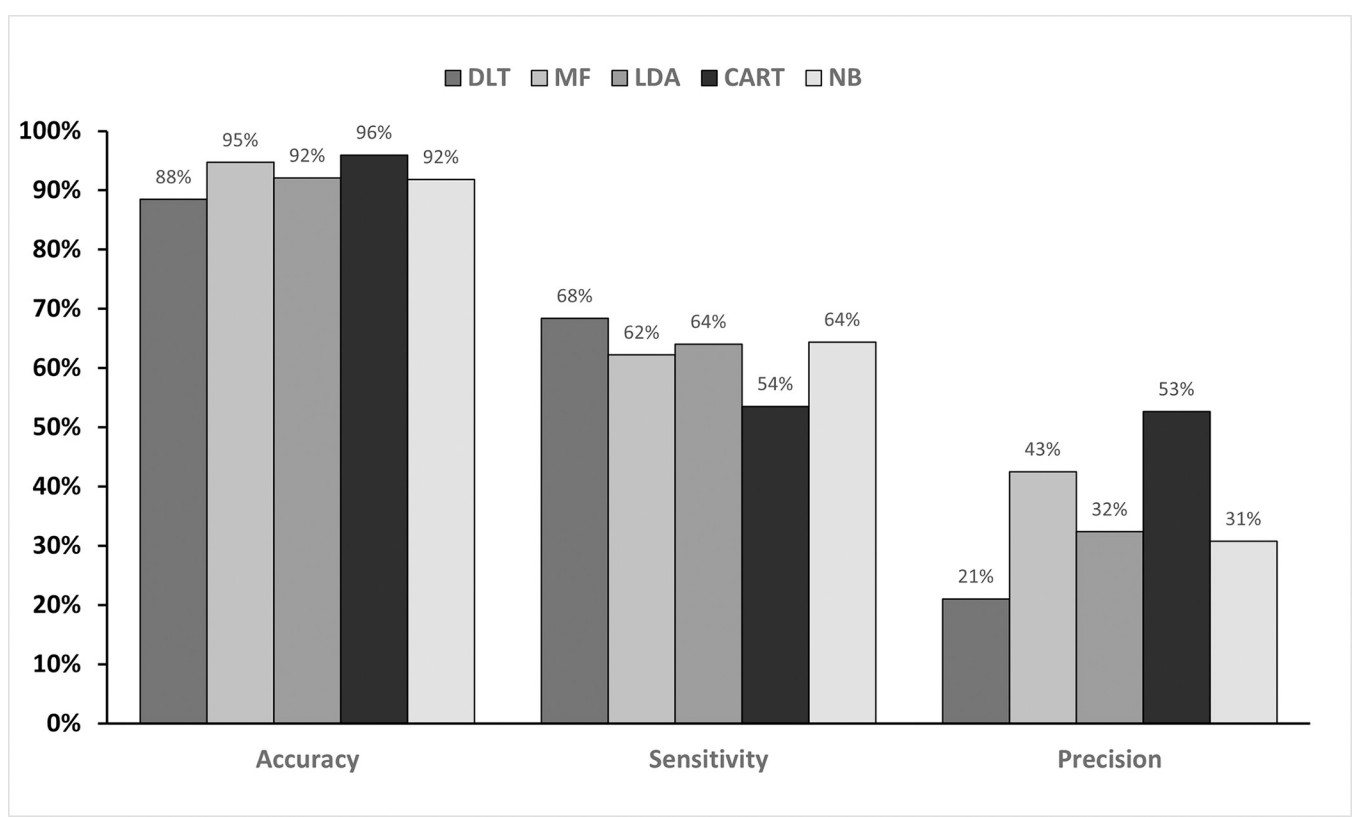

**Fig 4. Precision, sensitivity and accuracy of mold defect classification methods.**

## Visual comparison: Machine learning vs. morphological filtering

The visual results presented in Fig 6 suggest that the proposed mold defect classification method has enhanced the performance of the original DLT method. In samples with prominent stroke features, such as in P1 092, P1 107 and P1 279, implementing both classifiers successfully removed most of the stroke pixel groups. However, in P2 255, the strokes were not removed due to their potential association with circular defects where their edges meet. In samples P1 127 and P2 039, while the morphological filter retained large pixel groups at the top end of the samples, CART correctly classified these features as non-defects, highlighting its superior performance in classifying defects in complex images.

## Discussions

### Limitation of the hole filter

The hole filter, embedded as part of the set of rules of the morphological filter, effectively removed the outline features generated by the DLT method. However, due to the universal application of these rules, some actual defect features were also eliminated because they contained hole areas. This scenario is evident in the P2 046 sample, where KNN correctly retained features for classification as molds, while the morphological filter removed the pixel groups based on the established rules. Fig 7 illustrates the process within the morphological filter, showcasing the successful removal of outline strokes on the top left of the sample but inadvertently removing actual defects on the bottom left. In contrast, KKN does not encounter this issue, as depicted in Fig 6.

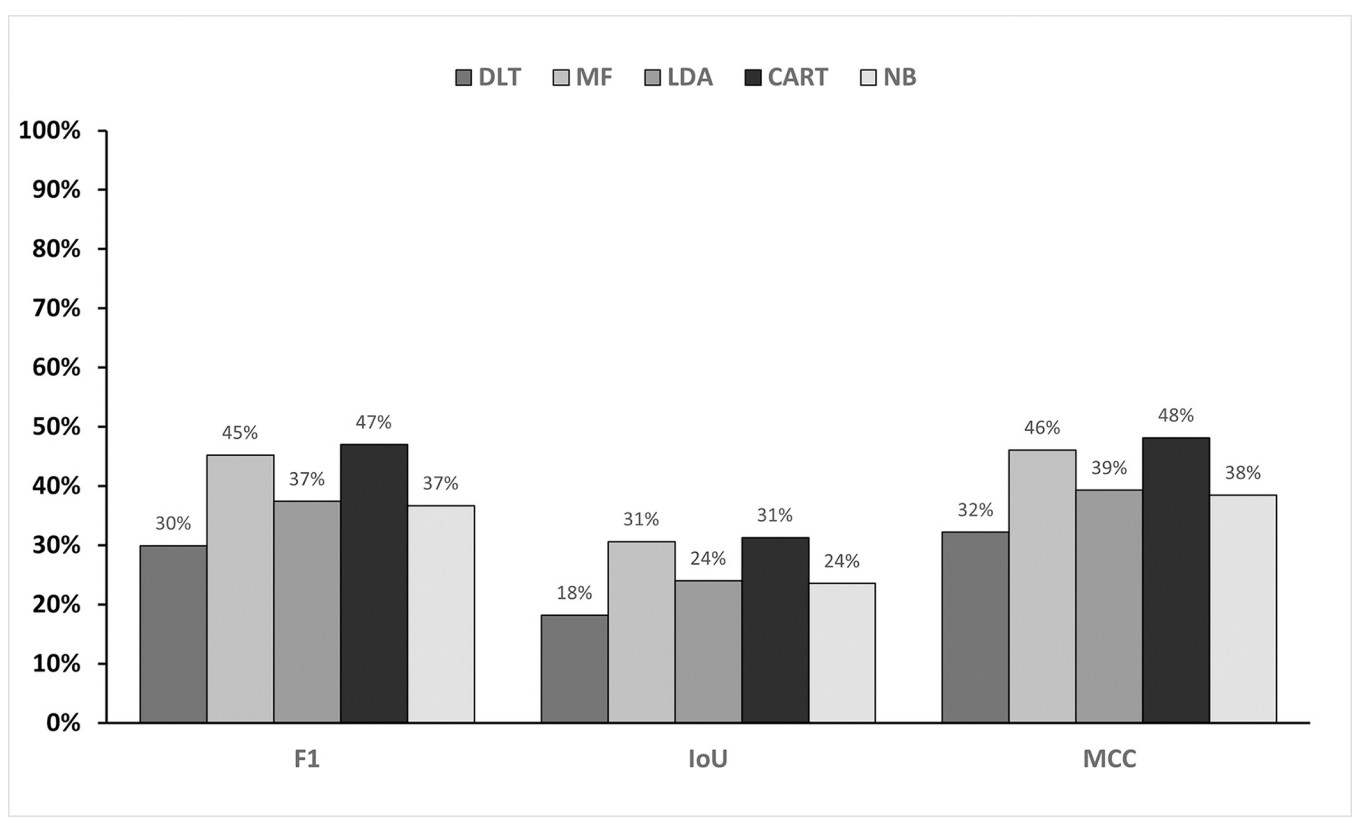

**Fig 5. IoU, F1-score and matthews correlation coefficient of mold defect classification methods.**

### LDA vs. CART for ML-based mold defect classification

The investigation into ML-based classifiers for mold detection in paintings yielded valuable insights, as illustrated in Fig 5. While CART emerged as the most accurate and precise method, LDA also has its own strength. LDA effectively identified mold defects in all ten test images and is more sensitive, demonstrating the ability to detect true positive cases (actual mold). This characteristic is crucial in the context of art preservation, where missing mold due to less sensitive classifier could lead to irreversible damage. Therefore, choosing between CART and LDA depends on specific applications and the relative importance of precision and sensitivity.

If minimizing false positives is more important, CART would be the preferred option due to its superior precision. This scenario might be relevant if any treatment for suspected mold area carries inherent risks. In situations where early detection of all potential molds is critical, LDA's high sensitivity becomes advantageous. This prioritization aligns with art preservation efforts, where missing mold altogether could be detrimental.

**Table 3. Statistical analysis of classification method accuracy.**

| Classification Methods | Mean | Standard Deviation | T-Statistic | p-value |
|---|---|---|---|---|
| Classification and Regression Trees (CART) | 0.9597 | 0.0169 | 3.5017 | 0.00997 |
| Morphological Filtering (MF) | 0.9470 | 0.0238 | 2.9471 | 0.02149 |
| Linear Discriminant Analysis (LDA) | 0.9209 | 0.0625 | 3.1340 | 0.01652 |
| Gaussian Naïve Bayes (NB) | 0.9184 | 0.0578 | 3.3754 | 0.01183 |

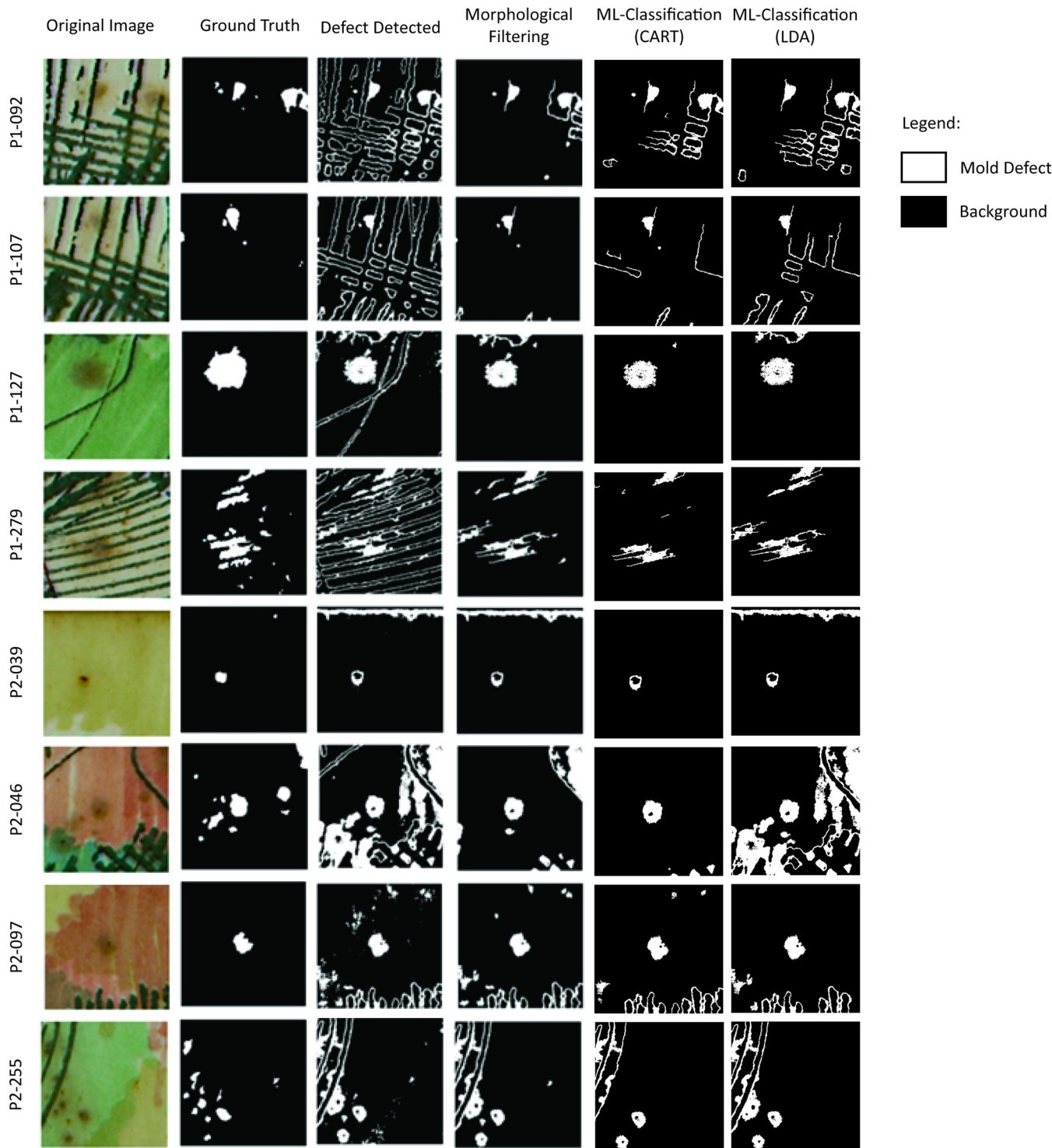

**Fig 6. Mold defect classification results for ML classification and morphological filtering.**

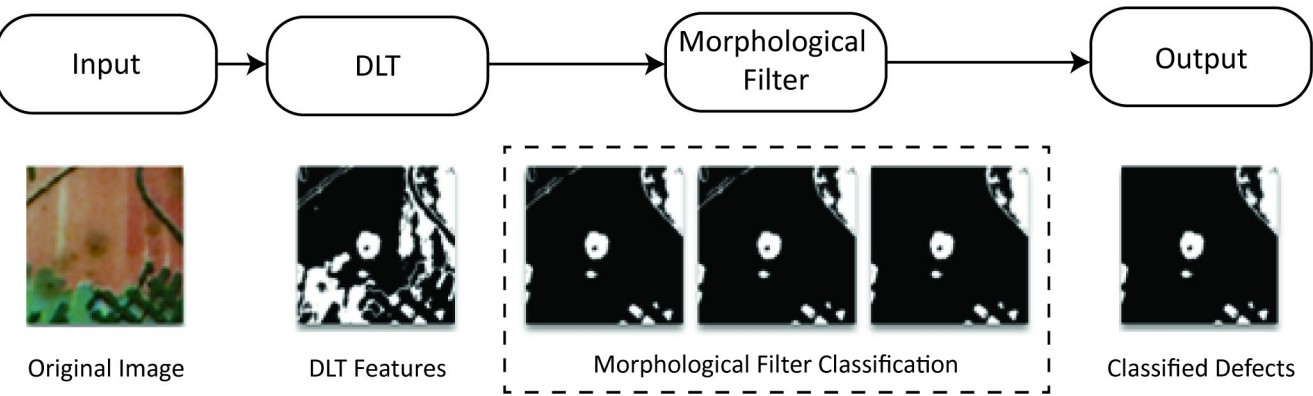

**Fig 7. Sample binary images produced using the DLT feature extraction method.**

## Dataset limitations

In general, the performance of both morphological filtering and machine learning (KNN) defect classification methods heavily relies on the feature extraction method. It is crucial for feature extraction algorithms to extract as many correct defective candidates as possible, but this often reduces the sensitivity of the algorithm. The DLT method has proven to be effective for feature extraction in identifying mold defects in painting. However, the resulting binary images have connected pixel groups, which reduces the mold defect classification accuracy.

To enhance classification accuracy, future work could consider the following approaches:

1. Replace feature extraction with features extracted by hidden layers in a Convolutional Neural Network with semantic segmentation. This approach has been successful in image classification tasks and is likely to be effective for defect classification in cultural heritage.

2. Utilize multiple feature extraction algorithms to generate more features that can be combined. This approach can capture more information from the images and improve accuracy of the classification model [10].

3. Address the issue of dataset class imbalance. This can be achieved by oversampling the minority classes or undersampling the majority classes. With a balanced dataset, the training accuracy of a machine learning model can be improved.

4. Apply data augmentation techniques such as rotation, flipping, and noise addition to artificially expand the dataset. There are also advanced techniques like GAN-based data generation to create synthetic dataset that has been proven to address imbalanced dataset issue [46].

5. Utilize hyperparameter tuning tailored to each specific machine learning model employed. For instance, when tuning a Random Forest, parameters such as the number of decision trees in the forest and the number of features considered at each node can be optimized. This ensures that each machine learning model is optimized for optimal performance comparison.

## Conclusions

This study has demonstrated the potential and feasibility of mold defect classifications in fine art paintings using both rule-based algorithms and machine learning. The Derivative Level Tresholding (DLT) method was employed for feature extraction and as a performance

benchmark for both approaches. The results revealed that the Linear Discriminant Analysis (LDA) machine learning model outperformed the other seven machine learning algorithms in the mold defect classification validation, achieving an accuracy of 96.7% and successfully detecting mold defects in all ten images in the testing dataset. However, when compared to morphological filtering as a mold defect classification method, the Classification and Regression Trees (CART)-based filtering demonstrated superior results, increasing precision by 32% to 53% while maintaining high accuracy (96%) comparable to morphological filtering method. These findings provide valuable insights into defect classification for mold defects on painted artworks and contribute to the broader field of surface defect detection in cultural heritage. The study highlights the potential of machine learning techniques to enhance the accuracy and reliability of mold detection in fine art conservation efforts.

## Author Contributions

**Conceptualization:** Hilman Nordin, Bushroa Abdul Razak, Norrima Mokhtar.

**Data curation:** Hilman Nordin.

**Formal analysis:** Hilman Nordin.

**Funding acquisition:** Bushroa Abdul Razak, Norrima Mokhtar, Adeel Mehmood.

**Investigation:** Hilman Nordin.

**Methodology:** Hilman Nordin, Norrima Mokhtar.

**Project administration:** Mohd Fadzil Jamaludin.

**Resources:** Bushroa Abdul Razak, Adeel Mehmood.

**Software:** Norrima Mokhtar.

**Supervision:** Bushroa Abdul Razak, Norrima Mokhtar.

**Validation:** Norrima Mokhtar, Mohd Fadzil Jamaludin, Adeel Mehmood.

**Visualization:** Hilman Nordin.

**Writing – original draft:** Hilman Nordin.

**Writing – review & editing:** Bushroa Abdul Razak, Norrima Mokhtar, Mohd Fadzil Jamaludin, Adeel Mehmood.

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
