## [Decision Letter · Decision Letter 0]

17 Sep 2024

PONE-D-24-29724Automated Mold Defects Classification in Paintings: A Comparison of Machine Learning and Rule-Based TechniquesPLOS ONE

Dear Dr. Mokhtar,

Thank you for submitting your manuscript to PLOS ONE. After careful consideration, we feel that it has merit but does not fully meet PLOS ONE’s publication criteria as it currently stands. Therefore, we invite you to submit a revised version of the manuscript that addresses the points raised during the review process.

We look forward to receiving your revised manuscript.

Kind regards,

Yile Chen, Ph.D. in Architecture

Academic Editor

PLOS ONE

Journal Requirements: When submitting your revision, we need you to address these additional requirements. 1. Please ensure that your manuscript meets PLOS ONE's style requirements, including those for file naming. The PLOS ONE style templates can be found at https://journals.plos.org/plosone/s/file?id=wjVg/PLOSOne_formatting_sample_main_body.pdf and https://journals.plos.org/plosone/s/file?id=ba62/PLOSOne_formatting_sample_title_authors_affiliations.pdf 2. When completing the data availability statement of the submission form, you indicated that you will make your data available on acceptance. We strongly recommend all authors decide on a data sharing plan before acceptance, as the process can be lengthy and hold up publication timelines. Please note that, though access restrictions are acceptable now, your entire data will need to be made freely accessible if your manuscript is accepted for publication. This policy applies to all data except where public deposition would breach compliance with the protocol approved by your research ethics board. If you are unable to adhere to our open data policy, please kindly revise your statement to explain your reasoning and we will seek the editor's input on an exemption. Please be assured that, once you have provided your new statement, the assessment of your exemption will not hold up the peer review process.

Reviewers' comments:

Reviewer's Responses to Questions

**Comments to the Author**

1. Is the manuscript technically sound, and do the data support the conclusions?

Reviewer #1: Yes

Reviewer #2: Yes

2. Has the statistical analysis been performed appropriately and rigorously? 

Reviewer #1: Yes

Reviewer #2: Yes

3. Have the authors made all data underlying the findings in their manuscript fully available?

Reviewer #1: Yes

Reviewer #2: Yes

4. Is the manuscript presented in an intelligible fashion and written in standard English?

Reviewer #1: Yes

Reviewer #2: Yes

5. Review Comments to the Author

Reviewer #1: 1. In Introduction, the last paragraph citations must be revised carefully.

2. In Related Works, I expected to see a comparison between previous models (techniques) according to advantages and disadvantages as minimum.

3. In Methodology, the first paragraph shouldn't be about the dataset. But it should be about the proposed model.

4. In Methodology, Table 1 caption should be rephrased.

5. In Results and Discussion, Table 2 must be sorted according to the most important evaluation metric/s.

6. A comparison between previous and proposed techniques should be considered in Results and Discussion section.

7. The small size of the dataset used may affect the accuracy of models. This must be considered in the future work.

Reviewer #2: Thank you for the opportunity to review this manuscript. The study presents a comparative analysis of rule-based morphological filtering and machine learning techniques for detecting and classifying mold defects in fine art paintings. The methodology is well-described and the results demonstrate the potential of these techniques for improving the accuracy and precision of mold defect detection in art conservation.

1. The statistical analysis, while generally sound, could be more rigorous. The authors should consider adding significance tests to compare the performance of the different methods and providing confidence intervals for the performance estimates. Hyperparameter tuning for the machine learning models could also be explored.

2. There are some minor typographical and grammatical errors throughout the manuscript that should be corrected in a revision.

Thank you.

6. PLOS authors have the option to publish the peer review history of their article (what does this mean?). If published, this will include your full peer review and any attached files.

Reviewer #1: **Yes: **Tamer Abdel Latif Ali

Reviewer #2: No

---

## [Author Response · Author response to Decision Letter 0]

4 Dec 2024

Reviewer 1 Comments

• Comment 1: In Introduction, the last paragraph citations must be revised carefully.

• Response 1: Thank you for the feedback. The last paragraph in the introduction containing the citations has been revised for clarity (see line 73). A concluding remark has also been added at the end of the paragraph.

• Comment 2: In Related Works, I expected to see a comparison between previous models (techniques) according to advantages and disadvantages as minimum.

• Response 2: Thank you for the feedback. Two paragraphs were added to explain the advantages and disadvantages of the techniques discussed in the paper (see line 102).

• Comment 3: In Methodology, the first paragraph shouldn't be about the dataset. But it should be about the proposed model.

• Response 3: Thank you for the feedback. The first paragraph has been revised to explain about the proposed model rather than the dataset (see line 130).

• Comment 4: In Methodology, Table 1 caption should be rephrased.

• Response 4: Thank you for your feedback. We have reviewed the caption for Table 1 and made the necessary revisions. The updated caption now reads (see line 172):

o “Distribution of Quantitative Mold Features used in the Classification Models”

• Comment 5: In Results and Discussion, Table 2 must be sorted according to the most important evaluation metric/s.

• Response 5: Thank you for your valuable feedback. We have revised Table 2 to sort results according to the most important evaluation metric, Detection. This provides a clearer and a more focused comparison of the different methods (see line 287).

• Comment 6: A comparison between previous and proposed techniques should be considered in Results and Discussion section.

• Response 6: We have carefully considered your suggestions and incorporated significance analysis to compare the previous and the proposed techniques in the Results and Discussion section (see line 309).

• Comment 7: The small size of the dataset used may affect the accuracy of models. This must be considered in the future work.

• Response 7: We appreciate the reviewer’s comment regarding the potential impact of dataset size on model accuracy. We acknowledge that a larger dataset could enhance our results. In future work, we plan to apply data augmentation techniques, explore the use of Convolutional Neural Network, using multiple feature extraction algorithms and addressing the issue of dataset class imbalance (see line 354).

Reviewer 2 Comments

• Comment 1: The statistical analysis, while generally sound, could be more rigorous. The authors should consider adding significance tests to compare the performance of the different methods and providing confidence intervals for the performance estimates. Hyperparameter tuning for the machine learning models could also be explored

• Response 1:

o We agree that strengthening the statistical analysis would enhance the robustness of our findings. We have incorporated the use of significance test in the form of paired t-test and reported the p-values to assess the significance of it in Table 3. We also have added another literature to the references to support the approach (see line 309).

o While we acknowledge that hyperparameter tuning can potentially improve model performance, we chose to use the machine learning model as it is to establish baseline comparison with rule-based method. In future studies, we plan to explore hyperparameter tuning to optimize our models and potentially achieve better results (see line 366). 

• Comment 2: There are some minor typographical and grammatical errors throughout the manuscript that should be corrected in a revision.

• Response 2: The manuscript has been sent for proofreading and typographical and grammatical errors throughout the manuscript has been addressed.

---

## [Decision Letter · Decision Letter 1]

20 Dec 2024

Automated Mold Defects Classification in Paintings: A Comparison of Machine Learning and Rule-Based Techniques

PONE-D-24-29724R1

Dear Dr. Mokhtar,

We’re pleased to inform you that your manuscript has been judged scientifically suitable for publication and will be formally accepted for publication once it meets all outstanding technical requirements.

Kind regards,

Yile Chen, Ph.D. in Architecture

Academic Editor

PLOS ONE

Additional Editor Comments (optional):

Reviewers' comments:

Reviewer's Responses to Questions

**Comments to the Author**

1. If the authors have adequately addressed your comments raised in a previous round of review and you feel that this manuscript is now acceptable for publication, you may indicate that here to bypass the “Comments to the Author” section, enter your conflict of interest statement in the “Confidential to Editor” section, and submit your "Accept" recommendation.

Reviewer #1: All comments have been addressed

Reviewer #2: All comments have been addressed

2. Is the manuscript technically sound, and do the data support the conclusions?

Reviewer #1: Yes

Reviewer #2: Yes

3. Has the statistical analysis been performed appropriately and rigorously? 

Reviewer #1: Yes

Reviewer #2: Yes

4. Have the authors made all data underlying the findings in their manuscript fully available?

Reviewer #1: Yes

Reviewer #2: Yes

5. Is the manuscript presented in an intelligible fashion and written in standard English?

Reviewer #1: Yes

Reviewer #2: Yes

6. Review Comments to the Author

Reviewer #1: (No Response)

Reviewer #2: I have reviewed the revised manuscript and am pleased to see that all previous comments have been thoroughly addressed. The addition of statistical analysis, improved organization, and enhanced discussion of limitations and future work have significantly strengthened the paper.

7. PLOS authors have the option to publish the peer review history of their article (what does this mean?). If published, this will include your full peer review and any attached files.

Reviewer #1: No

Reviewer #2: No

---

## [Editor Report · Acceptance letter]

14 Jan 2025

PONE-D-24-29724R1 

PLOS ONE

Dear Dr. Mehmood, 

I'm pleased to inform you that your manuscript has been deemed suitable for publication in PLOS ONE. Congratulations! Your manuscript is now being handed over to our production team.

Kind regards, 

on behalf of

Dr. Yile Chen 

Academic Editor

PLOS ONE
